# Risk factors for and outcomes of acute kidney injury in ward-based hospital trauma admissions: A retrospective cohort analysis

Omar Kiwan[1]☯, Elizabeth Finnimore[2,3]☯, Benjamin D. James[1,2], Paul W. Robinson[2], Mohammed Al-Kalbani[1], Becky Bonfield[4,5], Darren Green[1,2]*

1 University of Manchester, Manchester, United Kingdom, 2 Salford Royal Hospital, Salford, United Kingdom, 3 University of Salford, Salford, United Kingdom, 4 University of Southampton, Southampton, United Kingdom, 5 University Hospital Southampton NHS Foundation Trust, Southampton, United Kingdom

☯ These authors contributed equally to this work.
* darren.green@manchester.ac.uk

## Abstract

### Background

Guidelines on risk assessment for acute kidney injury (AKI) are generalised and may not adequately consider atypical presentations such as trauma. Older people are largely absent in past studies of AKI after trauma, meaning there is an evidence gap of trauma-associated AKI risk factors in older people.

### Methods

We undertook a retrospective analysis of 2,211 ward-level hospital trauma admissions during 2014–2022. We identified risk factors associated with AKI in people aged ≥65 years and <65 years, and established the clinical impact of AKI in older and younger trauma cases.

### Results

In those aged ≥65 years, parameters significantly associated with AKI were age, CKD, heart failure, infection, lower limb trauma. In people <65 years, the significant risk factors were age, CKD, liver disease, coronary disease, and pelvic trauma. In both age groups, AKI was associated with a greater risk of length of stay >14 days but not mortality.

### Conclusions

This study shows that risk factors for AKI in older trauma patients are comparable to those found in most guidelines for AKI risk assessment, with the addition of lower limb trauma. This factor could be considered as a useful adjunct in trauma AKI risk assessment tools to facilitate stratified care.

**Data availability statement:** All data files are available from the Harvard Dataverse (doi:https://doi.org/10.7910/DVN/9MLEGR).

**Funding:** The author(s) received no specific funding for this work.

**Competing interests:** I have read the journal's policy and the authors of this manuscript have the following competing interests: BB and DG are members of the UK Kidney Association AKI specialist interest group. BB chairs the education sub-group of the SIG and DG sits on the SIG governance group. This does not alter our adherence to PLOS ONE policies on sharing data and materials.

## Introduction

Acute kidney injury (AKI) is one of the most common complications during hospitalisation, affecting approximately 5–12% of patients on inpatient wards [1]. AKI is defined by KDIGO as either a relative increase in serum creatinine >50% within 7 days, an absolute increase of >26.5 mmol/L over 48 hours, or by the presence of oliguria [2]. It is associated with significant worsening of clinical outcomes including 23% inpatient mortality, extended hospital stay, and development of chronic kidney disease (CKD) [3–5].

Guidelines on the evaluation for risk of AKI in the inpatient setting highlight a need for proactive monitoring in high-risk patients, and indicate well-described factors; older age, cardiovascular comorbidities, the use of particular therapeutic agents, and certain acute circumstances such as sepsis and decompensated heart failure [6–7]. These risk parameters are not typically stratified according to presentation, with guidance being generalised to encompass all patient groups. This is recognised as a limitation of the current approach to AKI risk assessment, with the concern that scenarios in which higher AKI risk is present due to atypical mechanism of renal injury may not be identified [7].

Major trauma represents a potentially atypical AKI presentation [8]. AKI has been estimated to develop in up to half of severely injured patients and is a leading cause of organ failure and consequently death [9]. Many parameters observed to be associated with AKI in trauma cases are not found in most guidelines for risk assessment. These are generally factors which are specific to, or much more highly prevalent in trauma than in the general hospital population. Examples are history of blunt trauma or abdominal trauma, red cell transfusion, higher injury severity score, pre-hospital resuscitation, coagulopathy, and rhabdomyolysis [10–11].

Longer duration of an AKI episode is associated with greater harm in the setting of trauma, and early identification and management of AKI has been shown to significantly improve trauma patient outcomes [9,12]. There is consequently a clear rationale for proactive AKI risk assessment in the trauma setting. However, the absence of trauma-specific AKI risk parameters in general guidance means that trauma units aiming to establish protocols for AKI care face a lack of information specific to their patient population, restricting the potential for optimal AKI strategy and management. Furthermore, even those studies identifying parameters associated with AKI risk in trauma are limited in their scope, being almost exclusively based on level 1 trauma units (using American College of Surgeons verification criteria), dominated by critical care cohorts, and focused on younger patients (mean age is typically <50 years) [10,11]. This is despite patients aged 55 and older comprising over 40% of trauma cases with a doubling of trauma in older people reported between 2008–2017, and nearly 60% of trauma-related deaths [13]. In part, this likely relates to the exclusion of older persons from critical care settings.

The aims of this evaluation were therefore to determine whether the risk factors for AKI on a ward based trauma admissions unit (TAU) were reflective of those risk factors found in standard AKI guidelines, and whether trauma-specific AKI risk factors in this setting differ between older and younger patients. The reasoning was to help

understand whether the planned establishment of an AKI risk assessment tool in our trauma unit could be addressed using existing generic tools or whether trauma-specific factors would need to be included to achieve optimal AKI care.

## Materials and methods

Salford Royal Hospital is a large teaching hospital in the Northwest of England, United Kingdom. It includes a Major Trauma Centre for the Greater Manchester area (population 3.5 million), comprising specialist trauma care for spinal and neurosurgical patients, and providing care for all trauma except for cardiothoracic and vascular. The trauma centre includes a 28-bedded, ward-level TAU for both regional major trauma cases and care of trauma in older people with support from specialist orthogeriatric physicians. Admission to TAU requires the presence of multiple injuries.

This was an internal retrospective subgroup analysis of an existing dataset as part of the Hospital's long-standing AKI quality improvement collaborative project, previously described in detail elsewhere [14,15]. In accordance with the NHS Health Research Authority guidance for using patient data, ethical approval and individual patient consent was not required as no individual patient participants were recruited and the analysis used only an existing, fully anonymised clinical dataset without any individual patient identifiers being accessible. For the same reason the study was not registered as a clinical trial (clinical trial number: not applicable). For the present analysis, data were used relating to non-elective admissions to the ward-level TAU during the period 1st April 2014–31st December 2022. Data were accessed for analysis on 30th April 2024. Cases were excluded if aged <18 years, admitted electively, if unrelated to trauma, or if transferred from another ward following in-hospital trauma. This included step-downs from intensive care or transfers from theatre.

Data included were demographic features, ICD-10 codes for chronic comorbidities (heart failure I50, CKD N18.9, coronary artery disease I20-I25, diabetes mellitus E10-E11, liver disease K70-K77), ICD-10 codes for acute confounding illnesses (COVID-19 [U07, U08], infection [A00-B99], sepsis [A02.1, A22.7, A24.1, A26.7, A32.7, A40, A41, A42.7, A54.8, B37.7, R65.1, T81.4], rhabdomyolysis [M62.82]), main trauma site (determined by Emergency Department Diagnostic codes and triage notes), admission laboratory and physiological observation data (full list found in Table 1). Admission values for these parameters were selected to reflect the need for AKI risk assessment from the earliest stage of admission.

Main trauma site was determined by the body location of the primary injury which led to admission. A further group a was included with secondary priority trauma where there was an alternative admission reason, with trauma representing a secondary diagnosis.

AKI were identified by the presence of an ICD-10 code diagnosis N17 relating to each admission episode. Additional outcomes to AKI were inpatient death and total hospital length of stay >14 days.

Risk factors for AKI were identified using multivariable logistic regression analysis. All data were anonymised and analysed using SPSS version 29 (IBM). Five models were analysed: model A = acute and chronic comorbid factors; model B = admission laboratory and observational values; model C = trauma body sites; model D = all variables. Independent variables included in models A, B, C were *chosen a priori*. Model D included all variables from models A,B,C found to have a p value <0.1 on univariable analysis of their association with AKI. Model E was limited to patients admitted via the regional Major Trauma Pathway and included trauma-specific parameters collected in this cohort where available. These were injury severity score (ISS), mechanism of injury, trauma site, admission NEWS2 score, and Glasgow Coma Score. Mechanism of injury was sub-divided into the following categories: blast, blow(s), burn, crush, fall less than 2m, fall more than 2m, shooting, stabbing, vehicle incident/collision.

Hazard ratios for adverse outcomes in AKI (inpatient death or length of stay >14 days) were derived using logistic regression adjusted for age, gender and comorbid factors. All analyses were undertaken in the whole cohort and for subgroups of patients aged <65 years (younger) and ≥65 years (older). Missing values were handled by removal from analysis rather than inserting imputed values. These analyses allowed us to determine if AKI was independently associated with outcome in these groups and to compare the extent to which this is the case between groups.

**Table 1. Clinical characteristics of study population.**

| Feature | Whole population | ≥65 years | <65 years | AKI | No AKI |
|---|---|---|---|---|---|
| N~ | 2211 | 765 | 1443 | 117 | 2094 |
| Age (mean, SD) | 53.3±22.2 | 78.6±8.3 | 40.0±14.1 | 74.2±16.0 | 52.2±22.0 |
| Male Gender | 1472 (66.6%) | 372 (48.6%) | 1098 (76.0%) | 58 (49.6%) | 1414 (68%) |
| AKI | 117 (5.3%) | 89 (11.6%) | 28 (1.9%) | – | – |
| Length of stay (days) | 5 (0–218) | 10 (0–155) | 4 (0–218) | 17 (0–119) | 5 (0–218) |
| **Comorbidities n (%)** | | | | | |
| Chronic kidney disease | 94 (4.3%) | 83 (10.8%) | 11 (0.8%) | 28 (24%) | 66 (3%) |
| Heart Failure | 87 (3.9%) | 81 (10.6%) | 6 (0.4%) | 28 (24%) | 59 (3%) |
| Diabetes Mellitus | 206 (9.3%) | 138 (18.0%) | 68 (4.7%) | 28 (24%) | 178 (9%) |
| Sepsis | 23 (1.0%) | 16 (2.1%) | 7 (0.5%) | 9 (8%) | 14 (<1%) |
| Infection | 123 (5.6%) | 67 (8.8%) | 56 (3.9%) | 25 (21%) | 98 (5%) |
| COVID-19 | 83 (3.8%) | 48 (6.3%) | 35 (2.4%) | 5 (4%) | 78 (4%) |
| Coronary disease | 74 (3.4%) | 63 (8.2%) | 11 (0.8%) | 12 (10%) | 62 (3%) |
| Liver disease | 96 (4.3%) | 33 (4.3%) | 63 (4.4%) | 11 (9%) | 85 (4%) |
| **Laboratory and Observations (median – range)** | | | | | |
| Creatinine (mcmol/L) | 76 (29 - 689) | 76 (33 - 450) | 75 (29 - 689) | 90 (40-392) | 80 (29-689) |
| C-reactive protein (mg/L) | 34 (0 - 409) | 38 (0 - 380) | 31 (0 - 409) | 75 (4-342) | 56 (0-409) |
| Albumin (g/L) | 43 (12 - 56) | 42 (24 - 56) | 43 (12 - 56) | 40 (24-53) | 42 (12-56) |
| Haemoglobin (g/L) | 138 (56 - 185) | 135 (56 - 185) | 140 (67 - 185) | 128 (56-168) | 136 (57-185) |
| Urea (mmol/L) | 5.2 (0.3 - 38.4) | 5.6 (0.8 - 37.1) | 5 (0.3 - 38.4) | 7 (1.6-30.1) | 6 (0.3-38.4) |
| Potassium (mmol/L) | 4.1 (2.6 - 7.3) | 4.2 (2.7 - 7.3) | 4.1 (2.6 - 6.9) | 4.3 (2.9-5.7) | 4 (2.6-7.3) |
| **Main site of injury** | | | | | |
| Head & neck | 760 (34.4%) | 215 (28.1%) | 542 (37.6%) | 23 (20%) | 737 (35%) |
| Spine | 296 (13.4%) | 117 (15.3%) | 179 (12.4%) | 21 (18%) | 275 (13%) |
| Torso | 326 (14.7%) | 144 (18.8%) | 182 (12.6%) | 21 (18%) | 305 (15%) |
| Pelvis | 107 (4.8%) | 55 (7.2%) | 52 (3.6%) | 4 (3%) | 98 (5%) |
| Upper limb | 102 (4.6%) | 38 (5.0%) | 64 (4.4%) | 10 (9%) | 97 (5%) |
| Lower limb | 219 (9.9%) | 98 (12.8%) | 121 (8.4%) | 19 (16%) | 200 (10%) |
| Secondary priority | 402 (18.2%) | 98 (12.8%) | 303 (21.0%) | – | – |

~ 3 with unknown age. SD = standard deviation.

## Results

2,211 trauma patient admissions occurred during the period selected for analysis. The mean age was 53.3±22.2 years, 35% of patients were aged ≥65 years. 67% of admissions were male, 4% had existing chronic kidney disease (CKD), 4% had heart failure, 9% had diabetes. Of the 2,211 admissions, AKI occurred in 117 cases (5%). Other complications of admission were infection (6%), COVID (4%), and sepsis (1%). The most common trauma site was head and neck (34%) followed by thoracic and spinal injuries (15% and 13% respectively). A full list of clinical characteristics is found in Table 1, showing the whole cohort and descriptors specific to younger and older patients and in those with and without AKI. There were no coded episodes of rhabdomyolysis in the cohort.

People who suffered AKI were older 74.2±16.0 years versus 52.2±22.0 years, and more likely to be female (50% versus 32%). All major chronic co-morbidities were more common in AKI cases (Table 1).

AKI was more common in older people than those aged <65 years (12% versus 2%). AKI occurred in 39% of trauma admissions complicated by sepsis, the figure rising to 50% of older patient trauma cases complicated by sepsis. Of

existing chronic comorbid factors, AKI was most common in people with heart failure (32%). This compares to AKI in 4% of people without heart failure. AKI occurred in 29% of people with existing CKD, and 14% of those with diabetes. AKI was more common in older people in all comorbid subgroups except liver disease.

With respect to trauma site, AKI was most common in people suffering pelvic trauma (9%), with a comparable figure in lower limb trauma. In older patients, lower limb trauma cases had the highest incidence of AKI (18%), whereas for younger patients pelvic trauma had the highest incidence (8%). Head and neck trauma had the lowest overall incidence of AKI (3%). A graphical comparison of AKI incidence in different subgroups according to comorbidities and trauma sites is found in Fig 1, which also shows event rates for different age groups.

Missing values were limited to laboratory and observations parameters other than 3 patients without a documented age. Laboratory parameters had < 5% missingness other than C-reactive protein (34%) and albumin (22%). Observations were available from 76% of cases.

In the analyses of parameters associated with AKI, in model A (comorbidities), heart failure had the greatest association with AKI onset in older people (hazard ratio [HR] 4.1 [95% confidence 2.3–7.2], $p < 0.001$). By contrast, the HR in CKD was 3.0 (1.6–5.3), $p < 0.001$. In younger patients, both liver disease (HR 4.4 [1.5–13.1], $p = 0.007$) and coronary disease (HR 11.6 [1.9–68.6], $p = 0.007$) were significantly associated with AKI. CKD was the only long-term condition significantly associated with AKI in both younger and older patients.

In the analysis of admission laboratory and observations results (model B), no parameters were significant in the overall group, nor in either age group. These included admission serum creatinine, urea, C-reactive protein, albumin, haemoglobin, systolic blood pressure, diastolic blood pressure, heart rate, and respiratory rate.

In model C (trauma site), head and neck trauma was used as the reference group. In the overall cohort analysis, only upper limb trauma was not associated with a greater likelihood of AKI compared to head and neck. In older patients, lower limb and secondary priority trauma had a statistically significantly greater likelihood of AKI (HR 2.6 [1.2–5.3], $p = 0.008$). In younger trauma cases, only pelvic trauma showed statistically significantly greater risk (7.4 [2.0–27.3], $p = 0.002$). Spinal trauma achieved near significance in younger people (HR 3.1 [1.0–9.7], $p = 0.053$).

In the overall model of risk factors for AKI (model D), significant parameters differed between older and younger patients. In those aged ≥65 years, the significant parameters were age, CKD, heart failure, infection, lower limb trauma and secondary priority trauma. In people under 65 years, the significant risk factors were age, CKD, coronary disease, and pelvic trauma. Table 2 provides a full list of variables in all groups. The diversity of ethnicity in the study cohort was limited 90% of patients identified as white. All other specific ethnicities each represented <2% of the cohort. For this reason, analysis of ethnicity was changed post hoc to the risk of AKI in people who identify as any ethnicity other than white. This was not significant within the model.

Model E only included patients who were admitted via the major trauma pathway, amounting to 499 cases. 225 patients were aged 65 years and over and 274 were under 65 years. The variables assessed in model E were ISS, GCS, initial NEWS score and mechanism of injury. As was the case with model B, no parameter was statistically significant in this analysis in any group.

The impact of AKI on inpatient outcomes is shown in Table 3. Here, AKI was associated with an overall greater risk of length of stay >14 days of 1.7 (1.1–2.6), $p = 0.023$. The HR for longer length of stay with AKI was greater in the younger person subgroup (2.8 [1.2–7.0], $p = 0.023$) than in older people (1.5 [0.9–2.5], $p = 0.089$). Overall, mortality was numerically but not statistically more likely in AKI (HR 1.7 [0.9–3.3], $p = 0.116$). Again, the effect size was greater in younger people than older people.

## Discussion

We believe this to be the first study of AKI risk factors specific to ward-level trauma admissions, and the first to comprehensively evaluate AKI risk in older trauma patients.

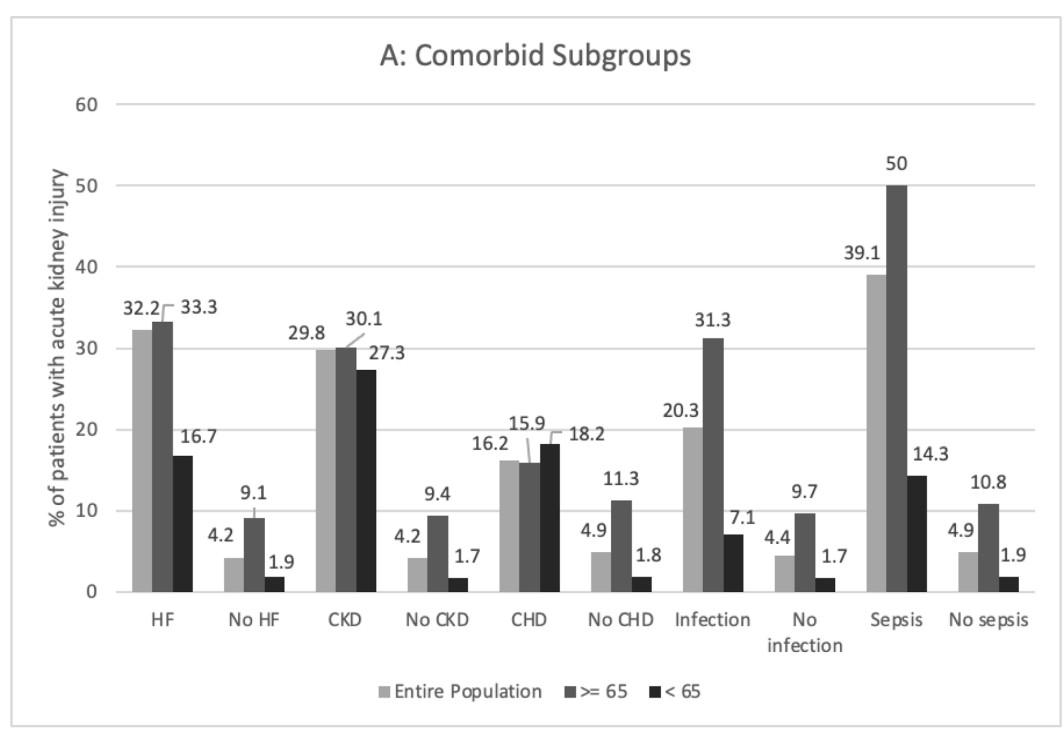

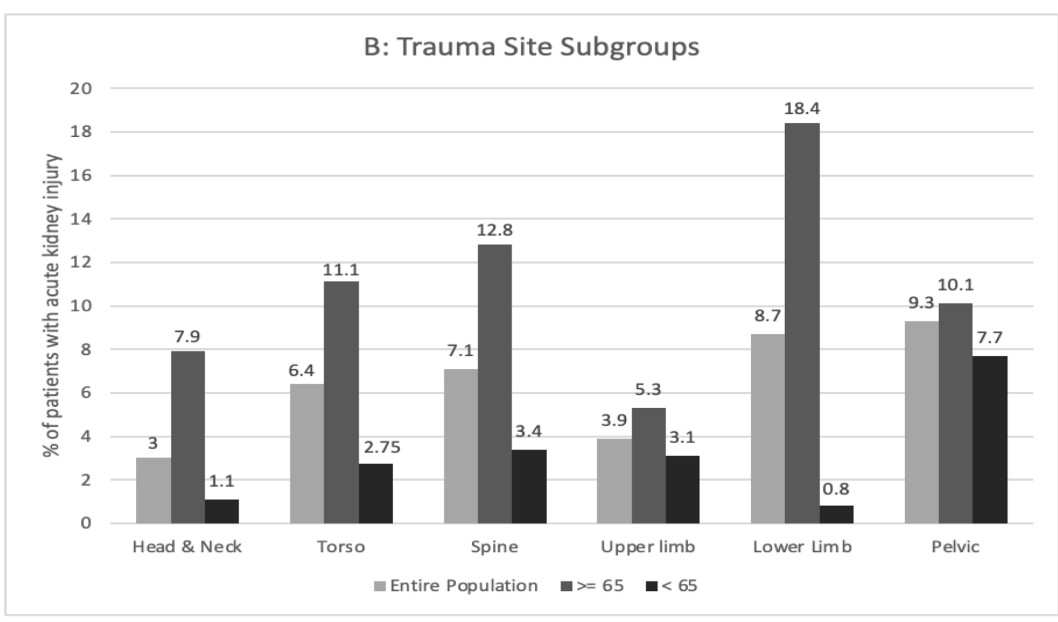

HF: heart failure, CKD: chronic kidney disease

**Fig 1. Likelihood of acute kidney injury in different population subgroups.** A=comorbid sub-groups, B=trauma site sub-groups. HF: heart failure, CKD: chronic kidney disease.

**Table 2. Multivariable models of factors associated with acute kidney injury in ward-level trauma admissions.**

| Feature | Whole population | | ≥65 years | | <65 years | |
|---|---|---|---|---|---|---|
| | HR (95% CI) | p | HR (95% CI) | p | HR (95% CI) | p |
| **Model A: comorbidities** | | | | | | |
| CKD | **4.9 (2.8-8.5)** | **<0.001** | **3.0 (1.6-5.3)** | **<0.001** | **15.1 (3.3-70.6)** | **<0.001** |
| Heart failure | **5.7 (3.2-10.1)** | **<0.001** | **4.1 (2.3-7.2)** | **<0.001** | 0.8 (0.0-16.4) | 0.890 |
| Diabetes mellitus | **2.1 (1.3-3.5)** | **0.003** | 1.3 (0.8-2.3) | 0.309 | **3.1 (1.0-9.3)** | **0.046** |
| Infection | **4.0 (2.3-6.9)** | **<0.001** | **3.7 (2.0-6.9)** | **<0.001** | 3.2 (0.9-10.2) | 0.054 |
| Coronary disease | 1.7 (0.8-3.6) | 0.168 | 0.9 (0.4-2.1) | 0.952 | **11.6 (1.9-68.6)** | **0.007** |
| Liver disease | 2.1 (0.9-4.3) | 0.054 | 1.2 (0.4-3.5) | 0.692 | **4.4 (1.5-13.1)** | **0.007** |
| **Model C: Trauma Site (reference group Head & Neck)** | | | | | | |
| Upper limb | 1.3 (0.4-3.9) | 0.627 | 0.6 (0.1-2.9) | 0.571 | 2.9 (0.6-14.6) | 0.201 |
| Spine | **2.4 (1.3-4.5)** | **0.004** | 1.7 (0.8-3.6) | 0.151 | 3.1 (0.98-9.7) | 0.053 |
| Torso | **2.2 (1.2-4.1)** | **0.010** | 1.5 (0.7-3.0) | 0.295 | 2.54 (0.8-8.4) | 0.130 |
| Pelvis | **3.3 (1.5-7.1)** | **0.002** | 1.4 (0.5-3.8) | 0.479 | **7.4 (2.0-27.3)** | **0.002** |
| Lower Limb | **3.0 (1.6-5.7)** | **<0.001** | **2.6 (1.3-5.3)** | **0.008** | 0.7 (0.1-6.2) | 0.786 |
| Secondary priority | 1.5 (0.9-3.0) | 0.141 | **2.1 (1.0-4.4)** | **0.049** | 1.2 (0.3-4.3) | 0.777 |
| **Model D: All variables*** | | | | | | |
| Age | **1.0 (1.0-1.1)** | **<0.001** | **1.0 (1.0-1.1)** | **0.023** | **1.1 (1.0-1.1)** | **0.009** |
| Ethnicity | 0.7 (0.4-1.3) | 0.241 | | X | | X |
| Female gender | 1.1 (0.7-1.7) | 0.759 | | X | | X |
| CKD | **2.7 (1.5-5.0)** | **0.001** | **2.4 (1.3-4.6)** | **0.008** | **11.2 (1.8-68.4)** | **0.009** |
| Infection | **3.9 (2.2-7.0)** | **<0.001** | **3.4 (1.7-6.8)** | **<0.001** | 3.7 (0.9-14.8) | 0.062 |
| Sepsis | **9.2 (3.2-26.0)** | **<0.001** | **6.2 (1.7-22.9)** | **0.007** | 5.8 (0.4-88.9) | 0.207 |
| Heart Failure | **3.1 (1.7-5.7)** | **<0.001** | **4.2 (2.3-7.7)** | **<0.001** | 0.6 (0.03-9.93) | 0.734 |
| Diabetes | **1.7 (1.0-2.8)** | **0.046** | 1.1 (0.6-2.2) | 0.722 | 2.2 (0.6-8.4) | 0.228 |
| Coronary disease | 1.3 (0.6-2.7) | 0.529 | | X | **8.7 (1.4-54.3)** | **0.021** |
| Liver disease | 2.2 (1.0-4.5) | **0.040** | | X | 2.1 (0.6-8.1) | 0.258 |
| Haemoglobin | | X | 1.0 (0.9-1.0) | 0.606 | 1.0 (0.9-1.0) | 0.181 |
| Albumin | | X | | X | 1.0 (0.9-1.1) | 0.708 |
| Trauma site | | 0.403 | | 0.272 | | 0.324 |
| Torso | 1.6 (0.8-3.1) | 0.171 | 1.5 (0.6-3.3) | 0.368 | 1.5 (0.4-6.0) | 0.570 |
| Spine | 1.8 (0.9-3.4) | 0.089 | 1.7 (0.8-3.8) | 0.180 | 1.8 (0.5-6.9) | 0.408 |
| Pelvis | 1.7 (0.7-4.0) | 0.240 | 1.0 (0.3-3.1) | **0.961** | **5.5 (1.1-26.4)** | **0.033** |
| Lower Limb | **2.0 (1.0-4.1)** | **0.045** | **2.4 (1.1-5.4)** | **0.029** | 0.7 (0.1-6.3) | 0.714 |
| Upper Limb | 1.3 (0.4-4.0) | 0.668 | 0.9 (0.2-4.1) | 0.834 | 2.2 (0.3-14.8) | 0.413 |
| Secondary | **2.1 (1.1-4.2)** | **0.027** | **3.1 (1.4-7.0)** | **0.006** | 1.0 (0.9-3.7) | 0.986 |

*Significant to p<0.1 on univariable analysis. X = insignificant in univariable analysis. Individual parameters reaching statistical significance (p<0.05) are highlighted in bold. Models B, E excluded from table as all variables non-significant.

The inclusion of older people may explain a higher rate of AKI in our study compared to previous retrospective analyses of trauma cohorts. Al-thani et al identified an AKI incidence of 0.8% in a Qatari National registry of 17,201 admissions to level 1 trauma units, the cohort age range being just 31–53 years [16]. Lai et al evaluated 14,504 admissions to a Thai level 1 trauma centre over a 6 year period and identified only 78 AKI. They determined an older age in AKI patients (62.9±21.0 and 52.7±19.2 years in people with and without AKI) but did not find any statistically significant comorbid risk factors for AKI on propensity score matching [17].

**Table 3. Adjusted hazard ratios for death and length of stay in people with acute kidney injury. LoS = length of stay. Individual parameters reaching statistical significance (p < 0.05) are highlighted in bold.**

| Feature | Whole population | | ≥65 years | | <65 years | |
|---|---|---|---|---|---|---|
| | HR (95% CI) | p | HR (95% CI) | p | HR (95% CI) | p |
| **All patients** | | | | | | |
| LoS > 14 days | 1.7 (1.1-2.6) | **0.023** | 1.5 (0.9-2.5) | 0.089 | 2.8 (1.2-7.0) | **0.023** |
| Inpatient death | 1.7 (0.9-3.3) | 0.116 | 1.5 (0.7-3.1) | 0.300 | 3.3 (0.7-16.3) | 0.135 |
| **Comorbidities: Heart failure** | | | | | | |
| LoS > 14 days | 1.9 (0.7-5.3) | 0.224 | 1.5 (0.5-4.4) | 0.475 | 0.0 (0 to-null) | 1.000 |
| Inpatient death | 1.2 (0.3-4.4) | 0.818 | 1.2 (0.3-4.5) | 0.807 | – | – |
| **Comorbidities: No heart failure** | | | | | | |
| LoS > 14 days | 1.8 (1.1-2.9) | **0.026** | 1.6 (0.9-3.1) | 0.127 | 2.3 (0.8-6.8) | 0.141 |
| Inpatient death | 2.0 (0.9-4.4) | 0.085 | 1.9 (0.7-5.3) | 0.169 | 4.1 (0.8-19.9) | 0.079 |
| **Comorbidities: Chronic kidney disease** | | | | | | |
| LoS > 14 days | 2.4 (0.8-6.9) | 0.109 | 4.1 (1.1-15.6) | **0.032** | 1.5 (0.5-40.0) | 0.828 |
| Inpatient death | 1.5 (0.3-8.3) | 0.657 | 1.6 (0.2-13.1) | 0.654 | – | – |
| **Comorbidities: No chronic kidney disease** | | | | | | |
| LoS > 14 days | 1.6 (1.0-2.6) | 0.073 | 1.2 (0.7-2.3) | 0.534 | 1.9 (0.6-6.2) | 0.275 |
| Inpatient death | 1.9 (0.9-3.9) | 0.076 | 1.7 (0.7-4.1) | 0.217 | 4.1 (0.8-19.9) | 0.079 |
| **Trauma site: Torso – 5** | | | | | | |
| Los > 14 days | 1.5 (0.5-4.3) | 0.495 | 1.4 (0.4-4.4) | 0.572 | 0.000 (0-null) | 0.999 |
| Inpatient death | 7.4 (0.6-86.3) | 0.111 | 12.4 (0.8-200.4) | 0.077 | 0 (0-null) | 0.999 |
| **Trauma site: Pelvis – 3** | | | | | | |
| Los > 14 days | 3.9 (1.2-12.7) | **0.027** | 2.1 (0.5-8.6) | 0.305 | 9.1 (1.4-61.1) | **0.023** |
| Inpatient death | 1.2 (0.3-5.8) | 0.815 | 0.5 (0.7-3.4) | 0.475 | 14.8 (1.2-182.4) | 0.036 |
| **Trauma site: Lower Limb – 2** | | | | | | |
| Los > 14 days | 3.5 (1.4-9.2) | **0.010** | 2.0 (0.6-6.6) | 0.243 | 0 (0-null) | 1.000 |
| Inpatient death | 2.3 (0.5-10.1) | 0.269 | 3.1 (0.6-16.3) | 0.186 | 0 (0-null) | 1.000 |

In subgroups of patients divided according to comorbid factors and trauma site, only one group showed a statistically significant greater risk of death after AKI. This was younger patients suffering pelvic trauma. Here the HR for death in AKI was 14.8 (1.2–182.4), p = 0.036 compared to patients without AKI.

Our study indicates that standard comorbid AKI risk factors are applicable to older trauma patients making generalised approaches to risk assessment valid in this setting. The only additional risk found specifically in older trauma cases is the presence of lower limb trauma. This should be included in AKI risk assessment tools for older trauma cases based on our findings. We hypothesise that this may reflect the high levels of frailty associated with falls and femoral fractures in older people, and a consequent AKI risk associated with frailty [18]. Lower limb trauma may also lead to greater muscle damage and resultant rhabdomyolysis. Creatine kinase may provide some utility in AKI risk assessment in trauma cases. However, in our study, admission routine laboratory data were not significant in AKI risk, and neither were admission observations or ISS. The lack of predictive association of admission observations or blood results with AKI suggest that the absence of physiological disturbance on patient arrival should not falsely reassure clinicians. Previous studies have shown an association between units of red cells transfused and increasing risk of AKI in younger trauma cases. We did not have transfusion data available for our analyses but found no association between admission haemoglobin and AKI.

In younger people, comorbid factors were less significant than in the older group, likely due to lower prevalence. This may represent type II error in the younger group in our cohort. Coronary disease and liver disease were noteworthy in being significantly associated with AKI in younger but not older trauma patients.

Heart failure is the most important comorbid factor in older person trauma-related AKI, more so even than CKD. We hypothesise this occurs due to concerns of overhydration in heart failure, and the possible inappropriate avoidance of intravenous fluids [19]. AKI in heart failure has significant implications beyond immediate care. There is overwhelming evidence of adverse long term disease outcomes after AKI in heart failure as AKI typically leads to suspension of necessary disease-modifying drugs. Specifically, renin-angiotensin-aldosterone system inhibitor (RAASi) drugs.

Infection is a significant biomarker of AKI risk in trauma cases, predominantly in older people. This is not specific to sepsis, although AKI rates are higher in the sepsis subgroups, as expected. The presence of infection after trauma is likely to indicate a complicated clinical scenario. AKI rates are higher in infection as a primary presentation so its effect in increasing risk in trauma cases is perhaps unsurprising.

Overall, we recommend approaching AKI risk and management in trauma patients following standard approaches as per available guidelines and recommendations. Regular measurement of serum creatinine paired with monitoring of hydration and urine output will provide indicators of AKI at the earliest stage. Some trauma-associated AKI will be unavoidable given that the primary insult which will cause the AKI has occurred before admission. However, the severity and duration of AKI can be reduced with early recognition and intervention [15]. Urgent imaging is indicated in major trauma cases, most often using computerised tomography. This will involve contrast agents which are associated with AKI. However, there is emerging recognition that this association is overstated, and does not implicitly indicate causation [20]. This is because people requiring contrast CT are at risk of AKI due to the reason a CT is required. Recommendations are increasingly moving towards highlighting that avoiding the use of contrast due to concerns of AKI risk may actually cause harm by missing diagnoses on imaging. Trauma admissions are an example of this, and it is important to state in local AKI guidelines that contrast should be used where indicated.

This study is not without limitations. Using ICD10 coding data means that our study is likely to have underestimated AKI rates. Epidemiological evidence shows that estimations of AKI incidence are much lower when using clinical identification or hospital coding data compared to laboratory data [21]. In addition, using AKI coding does not allow comparison of outcomes between AKI stages.

AKI onset is not time stamped: the study does not differentiate between community and hospital-acquired AKI. We hypothesise that AKI in the setting of trauma could broadly be either secondary to the initial injuries, or alternatively a downstream consequence of global stress, inflammation, and immobility. Sufficient time may not have elapsed for admission blood results to adequately represent the former, although in some circumstances traumatic events may significantly precede time of admission. We were also not able to include all trauma-linked parameters found to have association with AKI in previous studies. Specifically, we did not evaluate the impact of pre-hospital resuscitation, units of blood transfused or coagulopathy.

Analgesia is an important part of caring for people hospitalised for trauma. NSAIDs are a core component of effective pain relief but are themselves associated with worsening renal function. Our data capture did not include NSAID prescribing patterns but this is a topic deserving of consideration in future work on trauma-related AKI. Our hospital analgesic policy highlights the need to avoid NSAIDs in CKD and AKI, but whether the same is true for people at risk of AKI is worthy of exploration.

The low rate of AKI in younger people provides some reassurance but means that we are unable to comprehensively describe risk factors in this cohort as we are for older people. However, multiple other studies have described AKI in younger populations. Our study did not include some trauma-specific variables, such as units of blood transfused, which have been associated with AKI in previously published younger patient cohort studies.

Our study did not assign a cause to each AKI. The association of pelvic trauma with higher risk of AKI in younger patients raises the possibility of obstruction as a cause of AKI in the absence of specific AKI causality data. In our dataset there were no ICD10 coded episodes of urinary obstruction (N13.8, N13.9). OPCS4 coding for each admission identified 68 people who had non-elective urinary catheterisation during their admission. Overall, 3% of patients were

catheterised. The figure for pelvic trauma was 9%. Other primary trauma sites with higher catheterisation rates were spinal and lower limb trauma, both 5% of cases. Older people were more likely to be catheterised (6% of patients aged ≥80 years versus 2% of those aged <40 years). Whilst it can be hypothesised that there is a relationship between pelvic trauma and obstruction as the cause of AKI, other factors such as prolonged immobilisation may also be the reason for catheterisation.

## Conclusion

Risk factors for AKI in older trauma patients are comparable to those found in most guidelines for AKI risk assessment, with the addition of lower limb trauma. In addition, heart failure is seen as a dominant chronic comorbid factor and the role of infection in AKI risk is heightened. Few standard parameters are strongly predictive of AKI in younger trauma cases. Consideration of trauma-specific variables as identified in other studies may be more suitable for risk assessment in these patients. These differences between older and younger people should be taken into consideration when developing local AKI risk assessment tools stratified for ward-level trauma units.

## Author contributions

**Conceptualization:** Omar Kiwan, Elizabeth Finnimore, Paul W Robinson, Darren Green.

**Data curation:** Omar Kiwan, Elizabeth Finnimore, Paul W Robinson, Mohammed Al-Kalbani, Darren Green.

**Formal analysis:** Omar Kiwan, Benjamin D James, Mohammed Al-Kalbani, Darren Green.

**Investigation:** Elizabeth Finnimore, Paul W Robinson.

**Methodology:** Omar Kiwan, Elizabeth Finnimore, Benjamin D James, Paul W Robinson, Becky Bonfield, Darren Green.

**Project administration:** Elizabeth Finnimore, Benjamin D James, Becky Bonfield, Darren Green.

**Software:** Omar Kiwan, Benjamin D James, Paul W Robinson.

**Supervision:** Becky Bonfield, Darren Green.

**Validation:** Benjamin D James, Mohammed Al-Kalbani.

**Writing – original draft:** Omar Kiwan, Elizabeth Finnimore.

**Writing – review & editing:** Benjamin D James, Paul W Robinson, Mohammed Al-Kalbani, Becky Bonfield, Darren Green.

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
