## [Decision Letter · Decision Letter 0]

PONE-D-24-58849Risk factors for and outcomes of acute kidney injury in ward-based hospital trauma admissions: a retrospective cohort analysisPLOS ONE

Dear Dr. Green,

Thank you for submitting your manuscript to PLOS ONE. After careful consideration, we feel that it has merit but does not fully meet PLOS ONE’s publication criteria as it currently stands. Therefore, we invite you to submit a revised version of the manuscript that addresses the points raised during the review process.

We look forward to receiving your revised manuscript.

Kind regards,

Tarek Samy Abdelaziz, MD

Academic Editor

PLOS ONE

Journal Requirements:

“I have read the journal's policy and the authors of this manuscript have the following competing interests: BB and DG are members of the UK Kidney Association AKI specialist interest group. BB chairs the education sub-group of the SIG and DG sits on the SIG governance group.”

Reviewers' comments:

Reviewer's Responses to Questions

**Comments to the Author**

1. Is the manuscript technically sound, and do the data support the conclusions?

Reviewer #1: Yes

2. Has the statistical analysis been performed appropriately and rigorously? 

Reviewer #1: Yes

3. Have the authors made all data underlying the findings in their manuscript fully available?

Reviewer #1: Yes

4. Is the manuscript presented in an intelligible fashion and written in standard English?

Reviewer #1: Yes

5. Review Comments to the Author

Reviewer #1: Hello! i just reviewed the article entitled "Risk factors for and outcomes of acute kidney injury in ward-based hospital trauma admissions: a retrospective cohort analysis" by Omar Kiwan et al. The authors aimed to identify factors associated with AKI development in trauma patients, stratified by age in >65 and <65 years. The article is overall well-written, with a good flow, but before considering publishing the article, i have some suggestions:

1. You showed that pelvic trauma was more often associated with AKI in younger patients. Perhaps the pelvic trauma lead to obstructive AKI. Due to the fact that you do not have the AKI cause mentioned in the article (you present this in the limitations of the study), perhaps you can identify the number of patients with urinary catheters during the admission. Anyway, you should write a paragraph in the discussion section regarding this aspect.

2. Indeed, you mention that you identified AKI using the ICD score. Many studies showed that AKI is underdiagnosed using this score system, as many doctors are unaware of AKI. You should mention this in the discussion part also (e.g. of citation: "Holmes, Jennifer*; Rainer, Timothy; Geen, John; Roberts, Gethin; May, Kate*; Wilson, Nick*; Williams, John D.; Phillips, Aled O.; on behalf of the Welsh AKI Steering Group. Acute Kidney Injury in the Era of the AKI E-Alert. Clinical Journal of the American Society of Nephrology 11(12):p 2123-2131, December 2016. | DOI: 10.2215/CJN.05170516").

3. Do you have any data regarding the type of antalgic medication use? NSAID are know to increase the risk of AKI, especially in older or preexistent CKD patients. Being trauma patients, I consider that many of them required antalgic medication. If you can extract the number of days of NSAID and include this in the analysis of AKI occurrence, would increase the value of the results.

4. I consider you could remove this statement: "These are staples of care in heart failure, but only 50% of people who have experienced AKI will have RAASi reinstated by 6 months after AKI [20]. Readmission rates and long term survival are worse in heart failure patients who do not have these drugs reinitiated after AKI than those that do [21]." On the other hand you could talk about NSAID impact on heart failure patients and increased risk of AKI. Because you do not have follow-up after discharge, you should not present RAASi impact, unless you have data regarding RAASi use during hospitalization.

5. The discussion section seems short. You can increase it by 20-25% with no problem and you should introduce a paragraph or two regarding medical management of patients with trauma to reduce AKI incidence.

6. You can introduce in table 1 the LoS as a continuous variable.

7. Overall, a good article. Good luck!

6. PLOS authors have the option to publish the peer review history of their article (what does this mean? ). If published, this will include your full peer review and any attached files.

**Do you want your identity to be public for this peer review?** For information about this choice, including consent withdrawal, please see our Privacy Policy .

Reviewer #1: **Yes: ** MD, PhD, Lazar Chisavu, University of Medicine and Pharmacy "Victor Babes" from Timisoara, Romania, Nephrology Department

---

## [Author Response · Author response to Decision Letter 1]

29 Apr 2025

RESPONSE TO REVIEW

PONE-D-24-58849 - Risk factors for and outcomes of acute kidney injury in ward-based hospital trauma admissions: a retrospective cohort analysis

Dear colleagues,

We would like to thank the editorial team and the external reviewer for their time and effort in considering this manuscript and for providing comments that enhance the quality of the paper.

Please find below a point by point response to each comment. We have incorporated the feedback as much as possible in all cases, as all comments were fair and appreciated.

Best wishes

Darren Green

Editorial comments:

1.Please ensure that your manuscript meets PLOS ONE's style requirements, including those for file naming.

We have adjusted the manuscript formatting to meet the requirements as set out.

2. In the Competing Interests section please confirm that this does not alter your adherence to all PLOS ONE policies on sharing data and materials, by including the following statement: "This does not alter our adherence to PLOS ONE policies on sharing data and materials.”

The cover letter has been changed to include the above statement as requested.

3. We note that you have indicated that there are restrictions to data sharing for this study. PLOS only allows data to be available upon request if there are legal or ethical restrictions on sharing data publicly.

The study dataset can now be accessed without restrictions. We have uploaded the study dataset to the Harvard Dataverse: https://doi.org/10.7910/DVN/9MLEGR

Reviewers' comments:

1. You showed that pelvic trauma was more often associated with AKI in younger patients. Perhaps the pelvic trauma lead to obstructive AKI. Due to the fact that you do not have the AKI cause mentioned in the article (you present this in the limitations of the study), perhaps you can identify the number of patients with urinary catheters during the admission. Anyway, you should write a paragraph in the discussion section regarding this aspect.

The following paragraph has been added to the limitations section of the discussion, to address the point raised above.

Our study did not assign a cause to each AKI. The association of pelvic trauma with higher risk of AKI raises the possibility of obstruction as a cause of AKI in the absence of specific AKI causality data. In our dataset there were no ICD10 coded episodes of urinary obstruction (N13.8, N13.9). OPCS4 coding for each admission identified 68 people who had non-elective urinary catheterisation during their admission. Overall, 3% of patients were catheterised. The figure for pelvic trauma was 9%. Other primary trauma sites with higher catheterisation rates were spinal and lower limb trauma, both 5% of cases. Older people were more likely to be catheterised (6% of patients aged ≥80 years versus 2% of those aged <40 years). Whilst it can be hypothesised that there is a relationship between pelvic trauma and obstruction as the cause of AKI, other factors such as prolonged immobilisation may also be the reason for catheterisation.

2. Indeed, you mention that you identified AKI using the ICD score. Many studies showed that AKI is underdiagnosed using this score system, as many doctors are unaware of AKI. You should mention this in the discussion part also (e.g. of citation: "Holmes, Jennifer*; Rainer, Timothy; Geen, John; Roberts, Gethin; May, Kate*; Wilson, Nick*; Williams, John D.; Phillips, Aled O.; on behalf of the Welsh AKI Steering Group. Acute Kidney Injury in the Era of the AKI E-Alert. Clinical Journal of the American Society of Nephrology 11(12):p 2123-2131, December 2016. | DOI: 10.2215/CJN.05170516").

The citation you mention has been added to the study references. The discussion has been amended to provide more detail on this:

This study is not without limitations. Using ICD10 coding data means that our study is likely to have underestimated AKI rates. Epidemiological evidence shows that estimations of AKI incidence are much lower when using clinical identification or hospital coding data compared to laboratory data [21]. In addition, using AKI coding does not allow comparison of outcomes between AKI stages.

AKI onset is not time stamped: the study does not differentiate between community and hospital-acquired AKI.

3. Do you have any data regarding the type of antalgic medication use? NSAID are know to increase the risk of AKI, especially in older or preexistent CKD patients. Being trauma patients, I consider that many of them required antalgic medication. If you can extract the number of days of NSAID and include this in the analysis of AKI occurrence, would increase the value of the results.

Unfortunately our data capture does not include the ability to find robust / detailed records of analgesic prescriptions. We have added the following text to the discussion on this topic, which raises important considerations for AKI care / prevention in trauma.

Analgesia is an important part of caring for people hospitalised for trauma. NSAIDs are a core component of effective pain relief but are themselves associated with worsening renal function. Our data capture did not include NSAID prescribing patterns but this is a topic deserving of consideration in future work on trauma-related AKI. Our hospital analgesic policy highlights the need to avoid NSAIDs in CKD and AKI, but whether the same is true for people at risk of AKI is worthy of exploration.

4. I consider you could remove this statement: "These are staples of care in heart failure, but only 50% of people who have experienced AKI will have RAASi reinstated by 6 months after AKI [20]. Readmission rates and long term survival are worse in heart failure patients who do not have these drugs reinitiated after AKI than those that do [21]." On the other hand you could talk about NSAID impact on heart failure patients and increased risk of AKI. Because you do not have follow-up after discharge, you should not present RAASi impact, unless you have data regarding RAASi use during hospitalization.

We have removed this section as advised and included comment on NSAIDs as detailed in the response to point 3 above.

5. The discussion section seems short. You can increase it by 20-25% with no problem and you should introduce a paragraph or two regarding medical management of patients with trauma to reduce AKI incidence.

We have included the following paragraph on management of AKI in trauma patients, alongside the additional text / paragraphs as per responses to comments 1,2 and 3 above.

Overall, we recommend approaching AKI risk and management in trauma patients following standard approaches as per available guidelines and recommendations. Regular measurement of serum creatinine paired with monitoring of hydration and urine output will provide indicators of AKI at the earliest stage. Some trauma-associated AKI will be unavoidable given that the primary insult which will cause the AKI has occurred before admission. However, the severity and duration of AKI can be reduced with early recognition and intervention [15]. Urgent imaging is indicated in major trauma cases, most often using computerised tomography. This will involve contrast agents which are associated with AKI. However, there is emerging recognition that this association is overstated, and does not implicitly indicate causation [20]. This is because people requiring contrast CT are at risk of AKI due to the reason a CT is required. Recommendations are increasingly moving towards highlighting that avoiding the use of contrast due to concerns of AKI risk may actually cause harm by missing diagnoses on imaging. Trauma admissions are an example of this, and it is important to state in local AKI guidelines that contrast should be used where indicated.

6. You can introduce in table 1 the LoS as a continuous variable.

LoS has been added to the table shown as median (range) for each group, page 10.

7. Overall, a good article. Good luck!

Thank you for your kind words!

---

## [Decision Letter · Decision Letter 1]

Risk factors for and outcomes of acute kidney injury in ward-based hospital trauma admissions: a retrospective cohort analysis

PONE-D-24-58849R1

Dear Dr. Green,

We’re pleased to inform you that your manuscript has been judged scientifically suitable for publication and will be formally accepted for publication once it meets all outstanding technical requirements.

Kind regards,

Tarek Samy Abdelaziz, MD,FRCP

Academic Editor

PLOS ONE

Additional Editor Comments (optional):

Reviewers' comments:

Reviewer's Responses to Questions

**Comments to the Author**

1. If the authors have adequately addressed your comments raised in a previous round of review and you feel that this manuscript is now acceptable for publication, you may indicate that here to bypass the “Comments to the Author” section, enter your conflict of interest statement in the “Confidential to Editor” section, and submit your "Accept" recommendation.

Reviewer #1: All comments have been addressed

2. Is the manuscript technically sound, and do the data support the conclusions?

Reviewer #1: Yes

3. Has the statistical analysis been performed appropriately and rigorously? 

Reviewer #1: Yes

4. Have the authors made all data underlying the findings in their manuscript fully available?

Reviewer #1: Yes

5. Is the manuscript presented in an intelligible fashion and written in standard English?

Reviewer #1: Yes

6. Review Comments to the Author

Reviewer #1: The authors performed the required changes. I consider the article improved. From my point of view, it is ready for publication.

7. PLOS authors have the option to publish the peer review history of their article (what does this mean? ). If published, this will include your full peer review and any attached files.

**Do you want your identity to be public for this peer review?** For information about this choice, including consent withdrawal, please see our Privacy Policy .

Reviewer #1: **Yes: ** Lazar Chisavu

---

## [Editor Report · Acceptance letter]

PONE-D-24-58849R1

PLOS ONE

Dear Dr. Green,

I'm pleased to inform you that your manuscript has been deemed suitable for publication in PLOS ONE. Congratulations! Your manuscript is now being handed over to our production team.

Kind regards,

on behalf of

Professor Tarek Samy Abdelaziz

Academic Editor

PLOS ONE